# Lung Adenocarcinoma Diagnosed at a Younger Age Is Associated with Advanced Stage, Female Sex, and Ever-Smoker Status, in Patients Treated with Lung Resection

**DOI:** 10.3390/cancers15082395

**Published:** 2023-04-21

**Authors:** Tommaso A. Dragani, Thomas Muley, Marc A. Schneider, Sonja Kobinger, Martin Eichhorn, Hauke Winter, Hans Hoffmann, Mark Kriegsmann, Sara Noci, Matteo Incarbone, Davide Tosi, Sara Franzi, Francesca Colombo

**Affiliations:** 1Department of Research, Fondazione IRCCS Istituto Nazionale dei Tumori, 20133 Milan, Italy; 2Translational Research Unit (STF), Thoraxklinik, Heidelberg University Hospital, 69126 Heidelberg, Germany; 3Translational Lung Research Center (TLRC), German Center for Lung Research (DZL), 69120 Heidelberg, Germany; 4Department of Thoracic Surgery, Thoraxklinik, Heidelberg University Hospital, 69126 Heidelberg, Germany; 5Department of Thoracic Surgery, Klinikum Rechts der Isar, Technische Universität München, 81675 Munich, Germany; 6Institute of Pathology, Heidelberg University Hospital, 69120 Heidelberg, Germany; 7Department of Surgery, IRCCS Multimedica, 20099 Sesto San Giovanni, Italy; 8Thoracic Surgery and Lung Transplantation, Foundation IRCCS Ca’ Granda Ospedale Maggiore Policlinico, 20122 Milan, Italy; 9Institute for Biomedical Technologies, CNR, 20054 Segrate, Italy

**Keywords:** cigarette, latency, smoking

## Abstract

**Simple Summary:**

Since the factors influencing age at diagnosis of lung adenocarcinoma are unknown, in our study, we examined the relationships of age at diagnosis with smoking habit, clinical stage of disease, and sex in Italian and German patients with lung adenocarcinoma who underwent lung adenocarcinoma resection. Our results indicated that smoking habit, advanced clinical stage (more aggressive and larger tumour), and female sex were variables associated with younger age at diagnosis. This study provides new findings on the clinical variables influencing age at diagnosis of lung adenocarcinoma and paves the way for studies on the genetic and molecular mechanisms responsible for these associations.

**Abstract:**

To date, the factors which affect the age at diagnosis of lung adenocarcinoma are not fully understood. In our study, we examined the relationships of age at diagnosis with smoking, pathological stage, sex, and year of diagnosis in a discovery (n = 1694) and validation (n = 1384) series of lung adenocarcinoma patients who had undergone pulmonary resection at hospitals in the Milan area and at Thoraxklinik (Heidelberg), respectively. In the discovery series, younger age at diagnosis was associated with ever-smoker status (OR = 1.5, *p* = 0.0035) and advanced stage (taking stage I as reference: stage III OR = 1.4, *p* = 0.0067; stage IV OR = 1.7, *p* = 0.0080), whereas older age at diagnosis was associated with male sex (OR = 0.57, *p* < 0.001). Analysis in the validation series confirmed the ever versus never smokers’ association (OR = 2.9, *p* < 0.001), the association with highest stages (stage III versus stage I OR = 1.4, *p* = 0.0066; stage IV versus stage I OR = 2.0, *p* = 0.0022), and the male versus female sex association (OR = 0.78, *p* = 0.032). These data suggest the role of smoking in affecting the natural history of the disease. Moreover, aggressive tumours seem to have shorter latency from initiation to clinical detection. Finally, younger age at diagnosis is associated with the female sex, suggesting that hormonal status of young women confers risk to develop lung adenocarcinoma. Overall, this study provided novel findings on the mechanisms underlying age at diagnosis of lung adenocarcinoma.

## 1. Introduction

Cigarette smoking is the main risk factor for lung cancers, although this tumour might also develop in non-smokers [1,2]. Two recent epidemiological studies, from France, the USA and Japan, reported an increase in the number of affected patients among women and non-smokers [2,3]. Smoke habit remains the major cause of lung cancer, indeed, smokers have a ~20-fold higher risk of developing lung cancer than non-smokers, i.e., about one additional risk unit is conferred by each additional cigarette smoked per day [4,5]. Nevertheless, it is still not established whether smoking affects the natural history of lung cancer, for example, by accelerating the disease progression to a more advanced stage. The age at diagnosis of lung cancer shows wide variations, with some individuals being diagnosed in their thirties or forties and others developing the disease in old age [6,7,8]. Nagy-Mignotte et al., 2011, also reported evidence for an inverse relationship between the number of pack-years and age at diagnosis and, in that study, it was observed that quitting smoking delayed the age of diagnosis for both females and males [7]. 

Lung carcinogenesis is believed to involve a series of multiple molecular changes that unfold over several decades until the clinical diagnosis of cancer [9,10]. What controls the timing of these events is unknown. It is possible, for example, that this neoplasia has its own, inherent growth characteristics so that, once it has been induced, it develops and progresses independently of the inducing agent. This mechanism, in viral oncogenesis, has been described as a “hit and run” process in which a virus induces the malignant conversion of host cells (initiation), causing heritable changes that persist even after the loss of the viral genome, which is unnecessary for maintaining the malignant state [11]. Alternatively, lung carcinogenesis might proceed at a pace that depends on host factors, including the extent of exposure to carcinogens. 

Cigarette smoke, the main causative agent of lung cancer, might shorten the time to diagnosis by reducing the time required for each molecular oncogenic event to happen. Cigarette smoke contains over 60 known chemical carcinogens that damage DNA, leading to tumour induction and also acting in later phases of carcinogenesis [12]. In addition, exposure to cigarette smoke causes chronic inflammation and alterations in systemic immunity that may favour cancer promotion/progression [13]. 

To gain new insight into the mechanisms of lung carcinogenesis, in this study, we tested the effect of smoking habit on the age at diagnosis in lung adenocarcinoma patients. Moreover, we investigated the possibility that other parameters, such as sex and pathological stage, might be associated with the age at lung adenocarcinoma diagnosis.

## 2. Materials and Methods

### 2.1. Subjects

The discovery series in this study comprised 1694 lung adenocarcinoma patients who had undergone lung resection at one of three hospitals in the area around Milan, Italy (Fondazione IRCCS Istituto Nazionale dei Tumori, San Giuseppe Hospital, and Fondazione IRCCS Cà Granda Ospedale Maggiore Policlinico). Clinical data from these patients had been collected for previous studies [14,15,16]. The validation series comprised 1384 lung adenocarcinoma patients who had undergone lung resection at Thoraxklinik, Heidelberg, Germany. For the present study, we obtained data on the patients’ age at lung resection for adenocarcinoma diagnosis (hereafter called age at diagnosis), sex (male or female), self-declared smoking status (never or ever), pathological stage evaluated on surgical specimens, and year of diagnosis. The category of ever-smokers included both former smokers and current smokers. The pathological stage definitions were based on the 6th to 8th editions of TNM staging criteria for lung cancer by the Staging and Prognostic Factors Committee of the International Association for the Study of Lung Cancer (IASLC) [17,18,19].

### 2.2. Study Design and Statistical Analyses

In the present retrospective observational study, we tested the association between patients’ age at lung adenocarcinoma diagnosis and the following available patient information: sex, smoking status, pathological stage at diagnosis, and year of diagnosis. These four were the only variables of clinical interest for which full information was available for all the patients from the two series. Qualitative variables were described as number and percentages, whereas quantitative variables were described as median, range (minimum-maximum), 1st and 3rd quartiles (Q1 and Q3). First, the distributions of patients by clinical characteristic were compared between two series (Italian and German) using the chi-square test or Kruskal–Wallis test, for qualitative and quantitative variables, respectively. Then, we carried out a two-stage study, using a discovery and validation approach, where the regression analyses were carried out separately in a first cohort of Italian lung adenocarcinoma patients and then validated in a second patient series from Germany. A multivariable generalized linear regression model was used to test how sex, smoking habit, pathological stage, and year of diagnosis impact on age at diagnosis, as quantitative variable (primary outcome). We also performed logistic regression between the same variables, and age, as binary trait (above or below the median; secondary outcome). Linear and logistic models’ assumptions (i.e., normality of residual distribution, homoscedasticity for linear regression, outliers for logistic regression, and multicollinearity for both kind of models) were checked with the Performance package [20] in R environment. Pathological stage was coded either as a dummy variable, using stage I as reference, or as an ordered variable. The linear regression analyses produced beta whose values indicated that a given factor was independently associated with a younger (negative beta) or older (positive beta) age at diagnosis of lung adenocarcinoma. In the logistic regressions OR > 1 indicated that a given factor was independently associated with a younger age at diagnosis of lung adenocarcinoma. Statistical analyses were conducted in the R environment. A two-sided test *p* < 0.05 indicated statistical significance. 

## 3. Results

This study investigated 1694 lung adenocarcinoma patients in a discovery series and 1384 patients in a validation series (Table 1). In the discovery series, most patients were males (67.1%) and ever-smokers (82.9%); about half (55.6%) of patients were diagnosed with pathological stage I lung adenocarcinoma. Additionally, in the validation series, most patients were males (58.6%) and ever-smokers (85.4%); patients with pathological stage I disease comprised about one-third (34.5%) of cases. The distribution of age at diagnosis was different in the discovery and in the validation series (*p* < 0.001, Kruskal–Wallis test). In both discovery and validation series, the most represented age group was that of patients between 65 and 74 years. In the upper panel of Figure 1 we reported the age distributions in the discovery (A) and validation (B) series. Additionally, there was a statistically significant difference in the frequency of males and females in the two series, with a higher percentage of males in the discovery series (*p* < 0.001, chi-square test). Instead, the percentages of ever and never smokers were similar in the two series (*p* = 0.061, chi-square test). Finally, the distribution of patients by pathological stage differed between the two series, with a higher percentage of stage I patients in the discovery series and a higher percentage of stage III patients in the validation series (*p* < 0.001, chi-square test). The distributions of years of diagnosis were different in the discovery and in the validation series (*p* < 0.001, Kruskal–Wallis test). Patients from the discovery series were recruited in about four decades, with about 43% of them who underwent surgery in the 2001–2010 decade, about one third of patients were enrolled up to year 2000, and the remaining ones were diagnosed from 2011 to 2022. In the validation series, instead, patients were recruited in a shorter period, i.e., in two decades starting from 2000 (except for one patient in the ‘90s). In the lower panel of Figure 1, the distributions of the years of diagnosis in discovery (C) and validation (D) series were shown.

### Effects of Smoking Status, Pathological Stage, and Sex on Age at Diagnosis

The multivariable linear regression model (Table 2) showed that smoker status was an independent factor associated with younger age at diagnosis (ever-smokers vs. never-smokers: beta = −1.58, *p* = 0.0091). Pathological stage was also an independent factor associated with younger age at diagnosis: in particular, significant difference was observed at the highest stages (i.e., stage II versus stage I: beta = −1.17; *p* = 0.048; stage III versus stage I: beta = −2.27; *p* < 0.001; stage IV versus stage I: beta = −2.68; *p* = 0.0020). When we considered stage as an ordered variable, we observed a statistical significance for a linear trend (beta = −2.05; 95% CI = −3.23–−086; *p* < 0.001). Additionally, sex was associated with age at lung adenocarcinoma diagnosis, in particular, males were older than females at diagnosis (male versus female sex: beta = 2.46; SE = 0.50; *p* < 0.001). Finally, the year of diagnosis was significantly positively correlated with the age at diagnosis (beta = 0.29; *p* < 0.001). Since a non-normal distribution of residuals and heteroscedasticity (*p* < 0.001, both tests) was detected (Appendix A), we tested a different regression model.

A multivariable logistic model, where age was binarized in two groups of young (below the median age of 65) and old (above the median age) patients, was carried out and results similar to the previous ones were obtained (Table 3). No violations of model assumptions were detected (Appendix A). The risk of having a young age at diagnosis was higher in ever smokers than in never smokers (OR = 1.5, *p* = 0.0035), in females versus males (OR = 1.9, *p* < 0.001) and in stage III and IV patients as compared to stage I ones (OR = 1.4, *p* = 0.0067 and OR = 1.7, *p* = 0.0080, respectively). Again, when we considered stage as an ordered variable, we observed a statistical significance for a linear trend (OR = 1.49; 95% CI = 1.14–1.98; *p* = 0.0045). The risk of having a young age at diagnosis decrease with increasing years of diagnosis (OR = 0.94, *p* < 0.001).

In the validation series (Table 4), multivariable linear regression analysis showed that younger age at diagnosis was associated with the status of being an ever smoker (beta = −5.90, *p* < 0.001), having high pathological stage disease (stage III versus stage I beta = −2.51, *p* < 0.001; stage IV versus stage I beta = −3.42, *p* < 0.001), and being female (male versus female beta = 1.60, *p* = 0.0027). When we considered stage as an ordered variable, we observed a statistical significance for a linear trend (beta = −2.65; 95% CI = −4.0–−1.3; *p* < 0.001). No significant association with the year of diagnosis was observed. Additionally, for this linear regression model we detected a non-normal distribution of residuals (*p* < 0.001), whereas the homoscedasticity assumption was validated (*p* = 0.118; Appendix A). Likewise, we carried out a logistic regression with data from validation series.

As in the discovery series, results obtained in a multivariable logistic model, with age as a binary variable, were similar to those obtained with the linear regression. No violations of model assumptions were detected (Appendix A). In detail, a higher risk of having a lung adenocarcinoma diagnosis at young age (below the median age) was observed in ever smokers (OR = 2.9, *p* < 0.001), patients with increasingly high pathological stage (stage III versus stage I, OR = 1.4, *p* = 0.0066; stage IV versus stage I, OR = 2.0, *p* = 0.0022) and in females (OR = 1.3, *p* = 0.032; Table 5). Similarly, in the model with stage as an ordered variable, we observed a statistical significance for a linear trend (OR = 1.71; 95% CI = 1.27–2.33; *p* < 0.001). No significant association was observed with the year of diagnosis. 

## 4. Discussion

In this study, we carried out a multivariable analysis of age at diagnosis in an Italian discovery series of lung adenocarcinoma in which patients had undergone surgical resection of the tumours. We found that younger age at diagnosis was associated with ever-smoker status, higher pathological stage, female sex, and year of diagnosis. In an independent series of lung adenocarcinoma patients, from Germany, we validated these results, except for the association with the year of diagnosis.

Our two series of patients: discovery and validation, share the distribution of smokers and non-smokers, but they differ in the frequency of males and females, age at diagnosis, pathological stage of the tumour and year of diagnosis. In particular, regarding this latter difference, patients in the discovery series were enrolled in a wider period of time than the patients in the validation series. This might explain the significant association with year of diagnosis only in the discovery series and not in the validation one, possibly because this covariate is a confounding factor only in the Italian patient series. Anyway, both groups were treated with surgery for the same histotype, adenocarcinoma, and were characterized for the same clinical parameters.

In both the discovery and validation series, the status of being ever-smoker was associated with earlier age at diagnosis, supporting the role of smoking in modulating the age at diagnosis of lung adenocarcinoma. This finding might be explained by attributing to smoke habit the primary causative factor of lung cancer, and a secondary role in the intermediate phases of carcinogenesis that lead to frank lung cancer development. Indeed, a multistage model for lung carcinogenesis based on epidemiological data suggested that a relevant pathogenetic mechanism may involve smoking-induced lung tumour promotion, rather than tumour initiation [21]. This hypothesis is supported by experimental evidence of tumour promoter activity exerted by some compounds found in cigarette smoking. These results reject the hit-and-run hypothesis of causation of lung cancer, in agreement with the established effects of smoking cessation on the reduction in lung cancer risk [22]. 

A difference between smokers and non-smokers in the age at diagnosis of lung adenocarcinoma suggests that there are molecular differences in the disease in the two populations, which may depend on the causative agents. Studies on somatic mutations found that the molecular landscape of lung cancer differs between smokers and non-smokers [10,23]. The difference resides not only in the number of mutations, with smokers having a higher somatic mutation burden, but also in genes affected by the mutations. In particular, mutations in *EGFR* (epidermal growth factor receptor) and *ERBB2* (Erb-B2 receptor tyrosine kinase 2) are more frequent in non-smokers than in smokers, while *KRAS* (Kirsten rat sarcoma viral oncogene homologue) and *BRAF* (v-Raf murine sarcoma viral oncogene homologue B) mutations are more frequent in smokers [23,24]. Thus, distinct molecular pathways drive lung adenocarcinoma in smokers and non-smokers, leading several authors to suggest that lung cancers in these two populations are distinct pathological entities [25]. Recently, this has also been confirmed by a genomic landscape study of lung cancer in non-smokers [26]. Unfortunately, we did not have available mutational data for all the patients and, therefore, we could not test in our two series any association between mutational status and age or smoking.

A novel finding of our study is the association between younger age at diagnosis and higher pathological stage, seen in both series. This result was obtained in cases ranging from 29 to 89 years of age and the effect is increasingly higher with increasing pathological stage, in both the series. One other study reported an association between these variables, finding a higher proportion of stage I disease in very young patients (18–30 years) than in those aged 31–40 years from the Surveillance, Epidemiology, and End Results (SEER) database [6]. The narrow age range studied, and the heterogeneous characteristics of SEER patients can explain these discrepant results.

A possible explanation of having found a higher stage disease in younger patients is that tumours with greater ability to spread grow faster than less invasive tumours, despite the same histotype, leading them to be diagnosed at an earlier age. This interpretation contrasts with the paradigm of the continual, gradual increase in tumour malignancy through the sequential phases of carcinogenesis: initiation, promotion, and progression. It also contrasts with the multistage model of carcinogenesis by which the accumulation of several sequential somatic mutations over time is required for the development of a clinically evident tumour [27,28]. According to these models, it would be expectable that tumours diagnosed in older people have more time to accumulate mutations and, hence, should be more invasive than tumours diagnosed in younger patients; thus, our result was somehow unexpected. However, we do not know how many mutations and which ones are necessary for a normal cell to develop into a frank tumour. Indeed, the heterogeneity in somatic mutations in non-small-cell lung cancer suggests that this process is still unclear [29]. Additionally, lung adenocarcinomas in young patients seem to harbour a different set of somatic mutations with respect to older patients [30]. 

Finally, we observed an association between younger age at diagnosis and the female sex. This finding is in line with the well-known women’s higher risk of developing lung adenocarcinoma, than men, particularly among non-smokers [31]. In a very recent paper Xu et al., in 2022, reported that females represent a higher proportion of adenocarcinoma patients, compared to men, in non-smokers vs. smokers. This result might find a possible explanation in the genetic component. The authors identified a set of differentially expressed genes in lung cancer patients, associated with sex and non-smoking status. Particularly, they found MAPK/PI3K and ER signalling to be associated with adenocarcinoma differently in males and females and linked to a different prognosis [2]. In alternative to a genetic explanation, sex hormones might also play a role in women lung carcinogenesis: indeed, oestrogens are reported to be a risk factor for lung cancer in young women (as reviewed in [32]). 

Our results may have underestimated the effect of cigarette smoking in the anticipation of lung adenocarcinoma diagnosis; in fact, we included in the category of smokers both active smokers (regardless of the number of cigarettes) and former smokers (i.e., people who smoked for different periods and quit before the diagnosis of adenocarcinoma). In doing so, we diluted the effect of the variable cigarette smoking habit. We could not do otherwise because we did not have the quantitative and temporal information related to cigarette smoking in our patient series. Another limitation of our results may be that in the various birth cohorts, the types of cigarettes and tobacco predominantly smoked did not remain identical, producing different lung tissue inflammation or damage. Finally, in the more recent birth cohorts, the effects of the variables under study may have been underestimated, as there is an obvious lack of older individuals with whom to compare.

## 5. Conclusions

In conclusion, by analysing the effects of smoking, pathological stage, and sex on age at diagnosis, we found consistent evidence that lung adenocarcinomas detected at a young age are more common in ever-smokers, in females, and are more likely to present at an advanced stage. The association between younger age at diagnosis and higher clinical stage, i.e., more aggressive, and more advanced cancer, is a new finding of the present study and deserves to be validated in further independent clinical series. Future research should seek to identify somatic and germline alterations associated with early age at diagnosis and advanced clinical stage to elucidate the relationships between these clinically important variables and the underlying genetic and molecular mechanisms.

## Figures and Tables

**Figure 1 cancers-15-02395-f001:**
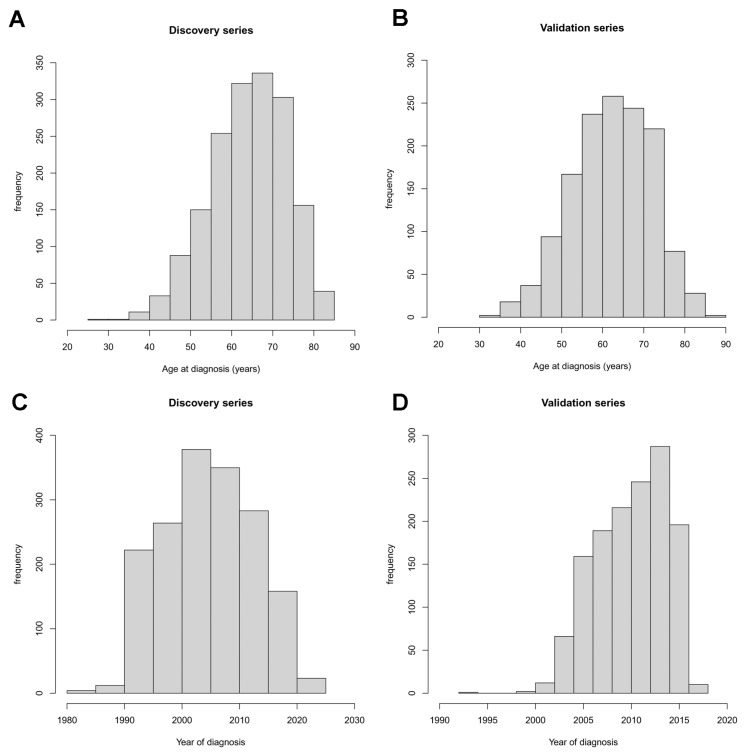
Distributions of age at diagnosis (upper panel) and year of diagnosis (lower panel) in the discovery (**A**,**C**) and validation (**B**,**D**) series.

**Table 1 cancers-15-02395-t001:** Clinical characteristics of lung adenocarcinoma patients in the discovery series and the validation series.

Factor	Discovery Series (n = 1694)	Validation Series (n = 1384)	*p*-Value
Age at diagnosis, years ^a^	65 (59–72), (29–85)	63 (56–70), (32–89)	<0.001 ^§^
Age group, n (%)			<0.001 ^#^
<55	251 (14.8)	280 (20.2)	
55–64	549 (32.4)	471 (34.0)	
65–74	665 (39.3)	498 (36.0)	
≥75	229 (13.5)	135 (9.75)	
Sex, n (%)			<0.001 ^#^
Female	557 (32.9)	572 (41.3)	
Male	1137 (67.1)	812 (58.7)	
Smoking habit, n (%)			0.065 ^#^
Never	290 (17.1)	203 (14.7)	
Ever	1404 (82.9)	1181 (85.3)	
Pathological stage, n (%)			<0.001 ^#^
I	942 (55.6)	478 (34.5)	
II	292 (17.2)	310 (22.4)	
III	341 (20.1)	491 (35.5)	
IV	119 (7.02)	105 (7.59)	
Year of diagnosis ^a^	2005 (2000–2011), (1981–2022)	2011 (1008–2014), (1992–2018)	<0.001 ^§^

^a^ Median (Q1–Q3 range), (min–max range), ^§^ Kruskal–Wallis test, ^#^ Chi square test.

**Table 2 cancers-15-02395-t002:** Discovery series: association of age at diagnosis of lung adenocarcinoma with smoking habit, pathological stage, sex, and year of diagnosis in 1694 patients.

Factor	Age at Diagnosis, Median (Q1–Q3), (Range)	Multivariable Glm ^a^	
		Beta (95% CI)	*p*-Value
Smoking habit			
Never	68 (59–73), (29–85)	1.0	
Ever	65 (59–71), (36–84)	−1.58 (−2.8–−0.39)	0.0091
Pathological stage			
I	66 (60–72), (36–85)	1.0	
II	66 (59–72), (29–84)	−1.17 (−2.3–−0.010)	0.048
III	64 (56–70), (38–84)	−2.27 (−3.4–−1.2)	<0.001
IV	61 (55–69), (36–84)	−2.68 (−4.4–−0.98)	0.0020
Sex			
Female	64 (56–71), (33–85)	1.0	
Male	66 (60–72), (29–84)	3.27 (2.3–4.2)	<0.001
Year of diagnosis		0.29 (0.24–0.35)	<0.001

^a^ glm, generalized linear model; CI, confidence interval.

**Table 3 cancers-15-02395-t003:** Results of the logistic regression with age at diagnosis, smoking habit, pathological stage, sex, and year of diagnosis in the discovery series.

Factor		OR	95% CI	*p*-Value
Smoking habit	never	1.0		
	ever	1.5	1.2–2.0	0.0035
Stage	I	1.0		
	II	1.2	0.91–1.6	0.21
	III	1.4	1.1–1.8	0.0067
	IV	1.7	1.2–2.6	0.0080
Sex	female	1.0		
	male	0.54	0.43–0.93	<0.001
Year of diagnosis		0.95	0.94–0.96	<0.001

OR, odds ratio of the risk of being diagnosed with a lung cancer at a young age (i.e., below the median age); CI, confidence interval.

**Table 4 cancers-15-02395-t004:** Validation series: association of age at diagnosis of lung adenocarcinoma with smoking habit, pathological stage and sex, in 1384 patients.

Factor	Age at Diagnosis, Median (Q1–Q3), (Range)	Multivariable Glm ^a^	
		Beta (95% CI)	*p*-Value
Smoking habit			
Never	69 (62–74), (32–89)	1.0	
Ever	62 (56–69), (33–85)	−5.90 (−7.3–−4.5)	<0.001
Pathological stage			
I	64 (57–71), (38–89)	1.0	
II	64 (56–70), (40–88)	−0.879 (−2.2–0.46)	0.20
III	62 (55–69), (32–82)	−2.51 (−3.7–−1.3)	<0.001
IV	60 (55–69), (33–81)	−3.42 (−5.4–−1.4)	<0.001
Sex			
Female	63 (55–70), (36–88)	1.0	
Male	63 (57–70), (32–89)	1.60 (0.56–2.6)	0.0027
Year of diagnosis		0.106 (−0.029–0.24)	0.12

^a^ glm, generalized linear model; CI, confidence interval.

**Table 5 cancers-15-02395-t005:** Results of the logistic regression with age at diagnosis, smoking habit, pathological stage, sex, and year of diagnosis in the validation series.

Factor		OR	95% CI	*p*-Value
Smoking habit	never	1.0		
	ever	2.9	2.1–4.0	<0.001
Stage	I	1.0		
	II	1.0	0.75–1.3	0.97
	III	1.4	1.1–1.8	0.0066
	IV	2.0	1.3–3.1	0.0022
Sex	female	1.0		
	male	0.78	0.62–0.98	0.032
Year of diagnosis		1.0	0.97–1.0	0.93

OR, odds ratio of the risk of being diagnosed with a lung cancer at a young age (i.e., below the median age); CI, confidence interval.

## Data Availability

Data sharing is not applicable to this article.

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
