# Peer review of "Lung Adenocarcinoma Diagnosed at a Younger Age Is Associated with Advanced Stage, Female Sex, and Ever-Smoker Status, in Patients Treated with Lung Resection"

_cancers, 2023, doi:10.3390/cancers15082395_

Round 1
Reviewer 1 Report
- Authors should provide information about what age they consider younger. It is also interesting to see the age distribution in each sample, and not just the median and mean age. Can be divided according to WHO guidelines or some other principle and analyzed in each subgroup by stage, smoking status and sex. It seems to me that this information will decorate the work.
- Was the EGFR, KRAS and BRAF mutation status not determined in the study groups? Associations with age may also be present in this case.
- Do the authors have more detailed information about the smoking experience? Those who quit smoking more than 10 years ago are significantly different from current smokers, for example.
Author Response
Response to Reviewer 1 Comments
Point 1: Authors should provide information about what age they consider younger. It is also interesting to see the age distribution in each sample, and not just the median and mean age. Can be divided according to WHO guidelines or some other principle and analyzed in each subgroup by stage, smoking status and sex. It seems to me that this information will decorate the work.
Response 1: To answer to this reviewer’s comment, in the revised manuscript we described the distribution of age in our series, in Table 1 (dividing patients in groups) and in Supplementary Figure 1. In addition, we reported the results of logistic regressions (Supplementary Table 1) in which age was considered as a binary variable (young vs. old patients, where the first were patients of age below the median and the latter were above the median). Linear and logistic models gave similar results.
Point 2: Was the EGFR, KRAS and BRAF mutation status not determined in the study groups? Associations with age may also be present in this case.
Response 2: Since all patients were surgically treated and, in addition, many of the patients were diagnosed several years ago, the mutational analysis was not performed. Mutational data were available only for a small subset of patients and thus, we could not evaluate their association with age. We discussed this limitation at page 7, lines 206-208.
Point 3: Do the authors have more detailed information about the smoking experience? Those who quit smoking more than 10 years ago are significantly different from current smokers, for example.
Response 3: Unfortunately, we did not have detailed information about smoking history of all the patients. We are aware of this limitation, and we had already discussed this point at page 7, lines 244-250.
Reviewer 2 Report
This is a well written manuscript.
Minor issues:
- Why did authors only classif stages I and >I. To have 1,2,3,4 stages can be very interesting to see if not only >1 but also for example stage 4 was more often in younger age.
- Table 2: this is usual to short p-values as <0.001 rather than something like 3.9 x 10-7, in both table and text.
- Not to have 'ex-smokers' and time from smoking stop is a big limitation of this study, as a big difference may be large between current and ex-smokers.
Author Response
Response to Reviewer 2 Comments
Point 1: Why did authors only classif stages I and >I. To have 1,2,3,4 stages can be very interesting to see if not only >1 but also for example stage 4 was more often in younger age.
Response 1: We thank the reviewer for this comment and, as suggested, we have added the information regarding the proportions of patients in each of the four stages, in Table 1. We also repeated the analyses using all the four categories of stage. Of note, we observed that the median age of patients with increasingly high stage was lower than patients with low stages and, as well, patients with highest stages had higher risk to have been diagnosed at younger age (below the median) than patients with low stage adenocarcinoma. These results have been reported in revised Tables 2 and 3 and in Supplementary Table 1.
Point 2: Table 2: this is usual to short p-values as <0.001 rather than something like 3.9 x 10-7, in both table and text.
Response 2: We agree with the reviewer that is common to show shorter P-values, but we think that reporting the exact of P-value is more informative, since smallest values indicates highest statistical significance. Thus, we did not change the displayed P-values.
Point 3: Not to have 'ex-smokers' and time from smoking stop is a big limitation of this study, as a big difference may be large between current and ex-smokers.
Response 3: We are aware of such limitation, but unfortunately, we did not have available full smoking history information for all the patients. We had already discussed this limitation at page 7, lines 244-250.
Reviewer 3 Report
In this article titled “Lung adenocarcinoma diagnosed at a younger age is associated with advanced stage, female sex, and ever-smoker status,” the authors have examined the relationships of age at diagnosis with smoking, stage, and sex, using lung adenocarcinoma samples from patients, who had undergone resection. The novel finding of this study is the association between younger age at diagnosis and higher pathological stage. To proceed with publication, some corrections are necessary -
Major comments
11) Line 46 – 48; Please re-word the statement “younger age at diagnosis is associated with the female sex, suggesting that women are genetically predisposed to develop lung adenocarcinoma.” The authors findings suggests that between male and female ever-smokers, females develop adenocarcinoma at a younger age – this does not indicate a “genetic predisposition”. At earlier age, females have higher levels and cycling of hormones, which could be a contributing factor. But, if there is a “genetic predisposition” due to female-sex you would see higher incidence of adenocarcinoma in females vs males, at every age group. Further, one would expect a gene mutation that is highly associated with the female-sex, if there is indeed a “genetic predisposition”.
In this age of misinformation, such statements are irresponsible.
22) What is the prevalent driver mutation of adenocarcinomas observed in younger ever-smoker patients of female-sex? KRAS mutation is frequently observed in adenocarcinoma patients with a history of smoking (as stated in lines 172–176), is there a difference in driver mutation associated with younger females vs older patients, male or female, with adenocarcinomas?
Author Response
Point 1: Line 46 – 48; Please re-word the statement “younger age at diagnosis is associated with the female sex, suggesting that women are genetically predisposed to develop lung adenocarcinoma.” The authors findings suggests that between male and female ever-smokers, females develop adenocarcinoma at a younger age – this does not indicate a “genetic predisposition”. At earlier age, females have higher levels and cycling of hormones, which could be a contributing factor. But, if there is a “genetic predisposition” due to female-sex you would see higher incidence of adenocarcinoma in females vs males, at every age group. Further, one would expect a gene mutation that is highly associated with the female-sex, if there is indeed a “genetic predisposition”.
In this age of misinformation, such statements are irresponsible.
Response 1: We thank the reviewer for this comment and we revised the sentence accordingly. In addition, we have briefly discussed the role of hormones in lung adenocarcinoma risk (page 7, lines 240-243).
Point 2: What is the prevalent driver mutation of adenocarcinomas observed in younger ever-smoker patients of female-sex? KRAS mutation is frequently observed in adenocarcinoma patients with a history of smoking (as stated in lines 172–176), is there a difference in driver mutation associated with younger females vs older patients, male or female, with adenocarcinomas?
Response 2: Unfortunately, we did not have available mutational data for all the patients. Indeed, since all patients were surgically treated and, many of the patients were diagnosed several years ago, the mutational analysis was not performed. In the revised manuscript we have added a sentence to discuss this limitation of our study (page 7, lines 206-208).
Round 2
Reviewer 1 Report
The authors answered my questions. I think that in its present form the article can be recommended for publication.
Author Response
We thank the reviewer for the recomendation for publication
Reviewer 3 Report
The recommended corrections have been made.
Author Response
No further corrections were requested.
To check the statistics in the manuscript, we invited additional reviewers.
Round 1
Reviewer 4 Report
GENERAL COMMENTS
While this paper is well written, its statistical analysis is extremely poor.
SPECIFIC COMMENTS
- In the Methods section, there is no explanation on how the study was designed, particularly regarding the statistical analysis.
- In the Methods section, no explanation is provided of the meaning of the discovery series that has been managed separately from the validation series.
- In studying the main factors influencing age at diagnosis, three variables were investigated (sex, smoking habit, and pathological stage); however, the Authors do not explain how these three variables were selected. More importantly, to optimize the selection of these variables, the Authors did not evaluate any statistical model of backwards elimination in which their number was reduced; likewise, no exploratory analysis (e.g. in terms of forward selection) was done to include other variables beyond the three variables mentioned above. A good reference on this topic can be found in: Chowdhury MZI, Turin TC. Variable selection strategies and its importance in clinical prediction modelling. Fam Med Com Health 2020;8:e000262. doi:10.1136/ fmch-2019-000262
- The Authors have ignored the statistical consequences that result from ordered categorical variables versus unordered categorical variables. A good reference on this topic can be found in: Moses LE, Emerson JD, Hosseini H. Analyzing data from ordered categories. N Engl J Med. 1984 Aug 16;311(7):442-8. doi: 10.1056/NEJM198408163110705. PMID: 6749191.
- There could be a serious error in the analysis because, in Table 1 (see the # symbol), some p-values were estimated based on the chi-square test as opposed to other p-values that were estimated by ANOVA. If the Authors confirm this point, the chi-square indicates that the analysis was a univariate analysis, not a multivariate analysis. This would greatly reduce the scientific value of the entire analysis reported in this paper.
- Reviewer 2 has correctly observed that reporting very small p-values in the order of 10-5 or 10-6 is meaningless; unfortunately, the Authors insist on the erroneous concept that relying on these values can provide useful information. I agree with Reviewer 2 that the Abstract should not be presented in the way the Authors have presented it.
- Depending on the requests made by the Reviewers, in doing their revision the Authors have converted binary variables into categorical variables ordered on three of more levels. The Authors have assumed that these two approaches are equivalent and interchangeable in terms of statistical modelling, but this assumption is incorrect: there are important statistical consequences related to this choice. In more detail: the variable “pathological stage” was initially handled as a binary variable (1 vs >1) and then converted into a multilevel variable based on 4 levels (1 vs 2 vs 3 vs 4); unfortunately, after this conversion, the Authors provided no information on whether this variable was considered to be ordered or unordered.
- An Appendix could be introduced in which the Authors test the statistical consequences of the two following choice: a) classifying each variable with three or more level as either an ordered categorical variable or an unordered categorical variable; b) exploring the effects of introducing a dummy variable defined as 0 or 1 for each level of a multi-level variables. For example, see: Polissar L, Diehr P. Regression analysis in health services research: the use of dummy variables. Med Care. 1982 Sep;20(9):959-66. doi: 10.1097/00005650-198209000-00008. PMID: 7121100.
Reviewer 5 Report
The main findings of the paper are centered around a statistical analysis, and a basic check reveals that this analysis lacks rigor. Indeed, ANOVA is a deep statistical test that needs several validation steps before drawing conclusions, and I don't see these steps in the paper. In particular, an ANOVA is valid if:
o The data's normal assumption holds (and we have no idea about this aspect for these data; no Shapiro-Wilk test, no descriptive graphics, nothing).
o The homoscedasticity: If the underlying variances are not equal, the ANOVA can not be applied. And again, there is no study on this in the paper.
These two significant missing validations cast doubt on the statistical plan's results.
For these reasons, my trust in the validation of the results of the statistical analysis is mitigated. I can not accept this paper.
Author Response
Point 1: The main findings of the paper are centered around a statistical analysis, and a basic check reveals that this analysis lacks rigor. Indeed, ANOVA is a deep statistical test that needs several validation steps before drawing conclusions, and I don't see these steps in the paper. In particular, an ANOVA is valid if:
• The data's normal assumption holds (and we have no idea about this aspect for these data; no Shapiro-Wilk test, no descriptive graphics, nothing).
• The homoscedasticity: If the underlying variances are not equal, the ANOVA can not be applied. And again, there is no study on this in the paper.
Response 1: The main results described in our paper are from linear and logistic regressions. In the previous version of the manuscript, we reported ANOVA results only for the comparison of age distribution between the two series. In the revised manuscript, since age at diagnosis and year at resection/diagnosis are not normally distributed, we compared them using the Kruskal-Wallis test and we reported these results in Table1. Additionally, in order to answer to reviewer 6’s comments we tested the assumptions of the linear and logistic models and reported the results of these tests in Supplementary Figures 1-4.
Reviewer 6 Report
The paper presents some wrong presentation of the methodology and the results. However, the used methodology is acceptable if the model hypotheses have been checked. At least looking the distribution of the age at diagnosis, I am confident about the distribution of the residuals of the model. Despite the necessity of reformatting, an important statistical/methodological point is the use of only 3 predictors. For a so huge cohort it is crucial that more variables are included. At least the year of diagnosis should be available. Finally the authors often mentioned gene and molecular mutations in references and introductions but nothing are presented in the results.
Major points:
- a) The statistical part is not well organise. After the population description:
- the authors have to defined the primary outcome: "The age at lung adenocarcinoma diagnosis". Then the secondary outcome which is the age at diagnosis as binary outcome.
- How are described qualitative and quantitative variables. For qualitative variables, while not written it is clearly number and percentage. However, for quantitative variable, it is not clear if it is median or mean for position parameter and Q1-Q3 or min-max for dispersion parameter. I suggest to provide Median, Q1, Q3, min and max.
- Univariate test used to compare distribution according to two groups. It is for me a bit strange to use an ANOVA test when a quantitative variable is compared according to only two groups, it is not false but more common to use a student t test.
- The multivariable linear model, the way assumption were checked. How were validated the assumptions of the model?
- The multivariable logistic model and how the assumption of the model were checked.
- The two stage study as discovery and validation cohort
- And finally the significant level, two-sides (?) tests and the software used
- b) It is crucial that studies variables are at time of diagnosis. Particularly the disease stage at diagnosis.Is it the case?
- c) Is there other potential factors? Only disease stage, sex and smoking habit is restrictive. What about the year of resection? Alcohol? Previous other cancer with chemotherapy and/or radiotherapy (before lung cancer)?
- d) The authors mentioned 3 previous published paper (14, 15, 16) for recuperation of the data. All these papers talk about genes and mutations and there importance. Why these data are not included?
- e) For the multivariable model, please provide the beta with 95%CI instead of SE.
- f) pvalues <0.001 have to be written as "<0.001" and not as "7.2*10^-6"
Minor points:
- a) The title does not correspond to the studies population. Indeed, the studied population is the one with lung resection for diagnosis of lung adenocarcinoma. Do all patients with lung adenocarcinoma have a lung resection? Not sure about that. I suggest clarify the title indicating "in patients treatment with lung resection".
- b) The authors mentioned sometimes "lung cancer" and sometimes "lung adenocarcinoma". The studies type of disease must be clearly identified and not generalised as all lung cancer are not adenocarcinoma
- c) On which criteria the age has been categorized?
- d) Line 129 "median age ...was higher ... ANOVA". The ANOVA doesn't test the median but the mean. If the authors wants to conclude on median they have to use Wilcoxon test that is more adequate. A more general term would be "the distribution".
- e) The distribution of the age at diagnosis is only on the supplementary, I suggest to include it in the manuscript.
Round 2
Reviewer 4 Report
The study has been improved considerably after the second revision.
In my view, three points need to be further addressed:
- The Methods section should preferably contain three sections: 2.1 Study design, 2.2 Subjects, 2.3 Statistical analyses.
- In their models, the Authors evaluated three variables of clinical interest for which full information was available for all the patients from the two series. This should be mentioned somewhere, e.g. in the Study design or in the Discussion.
- A final check by a statistician can be useful to confirm that the analysis is sound.
There seems to be a technical issue with the enclosed files because the Supplementary material (i.e, Supplementary Table 1, Supplementary Table 2, and Supplementary Figure 1) is not available in the revised manuscript.
MINOR COMMENTS
-Line 148: replace “significance” with “significant”.
Reviewer 6 Report
Authors answered my comments clearly and modified the manuscript accordingly.
Round 3
Reviewer 4 Report
Table 1, age at diagnosis, yearsa: two ranges are presented in two separate parentheses, e.g., 65 (59 – 72), (29 – 85) for the discovery series and 63 (56 – 70), (32 – 89) for the validation series. Please, explain more clearly: between 59-72 vs 29-85, which is the correct range? Likewise, which is the correct range between 56-70 vs 32-89? Presumably, 59-72 and 56-70 are correct, and so 29-85 and 32-89 should consequently been deleted. I wish to stress that this need for clarification does not refer to how the text is written, but to how the revised text has been explained.